# Explainable machine learning approach for cancer prediction through binarilization of RNA sequencing data

**Tianjie Chen, Md Faisal Kabir**  *

Department of Computer Science, Pennsylvania State University Harrisburg, Middletown, Pennsylvania, United States of America

* mpk5904@psu.edu

**Data Availability Statement:** The data underlying the results presented in the study are available from https://www.kaggle.com/datasets/tianjiechen/tcga-rna-datasets/data.

## Abstract

In recent years, researchers have proven the effectiveness and speediness of machine learning-based cancer diagnosis models. However, it is difficult to explain the results generated by machine learning models, especially ones that utilized complex high-dimensional data like RNA sequencing data. In this study, we propose the binarization technique as a novel way to treat RNA sequencing data and used it to construct explainable cancer prediction models. We tested our proposed data processing technique on five different models, namely neural network, random forest, xgboost, support vector machine, and decision tree, using four cancer datasets collected from the National Cancer Institute Genomic Data Commons. Since our datasets are imbalanced, we evaluated the performance of all models using metrics designed for imbalance performance like geometric mean, Matthews correlation coefficient, F-Measure, and area under the receiver operating characteristic curve. Our approach showed comparative performance while relying on less features. Additionally, we demonstrated that data binarization offers higher explainability by revealing how each feature affects the prediction. These results demonstrate the potential of data binarization technique in improving the performance and explainability of RNA sequencing based cancer prediction models.

## Introduction

Machine learning (ML) models have been used for cancer research for almost 40 years. In the past, researchers primarily focused on using clinical and demographic data to individual's risk of developing cancer [1]. Recent advancements in genomic and computational technology has enabled researchers to study cancer more thoroughly and develop new models for cancer prediction and survival analysis [2–5].

One proven way to study cancer using computational and ML-based methods is by analyzing peptides, specifically anti-cancer peptides (ACPs) data. Because of their low toxicity and greater efficacy, ACPs have recently attracted researchers' interests as a promising therapeutic agent for cancer treatment. However, efficient identification of ACPs is still a challenge. To

**Funding:** The author(s) received no specific funding for this work.

**Competing interests:** The authors have declared that no competing interests exist.

address this issue, researchers have proposed multiple ML-assisted tools for prediction of ACPs. Some early examples of utilizing ML models such as support vector machine (SVM) to identify ACPs are Chou's pseudo-amino acid composition (PseAAC) and sequence-based identification of anticancer peptides (iACP) [6, 7]. These methods paved the foundation for approaches that incorporated more advanced ML techniques like feature selection, dimensionality reduction, ensemble learning, and genetic algorithm. To further improve the performance of iACP method, a novel model called iACP-GAEnscC was developed based on ensemble learning and evolutionary genetic algorithm [8]. Dimensionality reduction techniques like principal component analysis (PCA) was also used to develop effective models like cACP for prediction of ACPs [9]. In particular, the cACP model was further developed into cACP-2LFS model, which utilized a two-level feature selection method to improve performance on existing models, and cACP-DeepGram model, which achieved better performance by incorporating a FastText-based word embedding strategy to represent each peptide sample [10, 11].

Another proven way is to utilize patients' RNA sequencing (RNA-seq) data. However, using RNA-seq data for cancer prediction poses a challenge to researchers because of their high dimensionality, complexity, and redundancy, which could lead to decrease in accuracy and efficency [12]. To combat this issue, researchers utilized dimensionality reduction and feature selection methods, such as univariate feature selection [13, 14], stepwise feature selection [15], PCA [16], autoencoder [17–19], and hybird approaches [20]. However, both feature selection and dimensionality reduction have some hard-to-fix drawbacks. Because feature selection functions by extracting a subset of features that is more related to the label, it ignores the inter-relationships between features [21]. On the other hand, interpreting new data of a lower dimension generated by dimensionality reduction techniques like PCA and autoencoder is difficult because one new feature could correlate to multiple original features [22].

Interpreting a ML model has always been a difficult task. Fundamentally, ML models can be divided into two groups: white box models and black box models. White box models like decision tree and logistic regression are easiler to interpret because they have built-in feature importance that explains how each feature contributes to predictions [23]. Black box models, on the other hand, must rely on post-hoc explanation approaches. A few popular post-hoc explanation techniques are Local interpretable model-agnostic explanations (LIME) [24], SHapley Additive exPlanations (SHAP) [25], saliency map, and counterfactual explanations. Although existing peptides and RNA-based models have shown encourging performances in producing accurate predictions, they are limited in terms of interpretability. Therefore, researchers have developed several models to address this issue. For peptides-based models, both white box and post-hoc explanation techniques were used. White box models like ACPred used rule extraction on random forest models to extract decision rules [26]. On the other hand, post-hoc explanation like LIME and SHAP are the preferred choice for explaining more complexed models that are based on neural networks or ensemble learning like iAFPs-EnC-GA, AIPs-SnTCN, and ACPred-BMF [27–29]. For RNA-based models, most studies chose post-hoc explanations like SHAP because of the complexity of the algorithms used [30, 31]. Despite having many choices, interpreting models that use continuous features is still difficult. For continuous features, most explanation techniques that generate feature importance scores only shows the names of importance features and how each of them contributes to predictions in terms of importance-like scores. This kind of interpretation is not meaningful as users would not understand how each feature, in terms of its original value, contributes to predictions. To combat this issue, researchers proved that, by binarilizing continuous features, interpreting ML models become much easier as each feature is meaningful [32].

Data binarilization has already been proven to be able to increase the interpretability of ML models while maintaining predictive accuracy. Similar approach has been used to identify

ACPs and demonstrated its effectiveness [33]. However, this technique has never been applied to RNA-seq data before. Therefore, we propose a data binarilization-based approach to increase the performance and interpretability of ML models used for cancer prediction.

The contributions of this study are:

- Data binarilization technique is proposed for processing RNA-seq data.

- Multiple models were constructed and tested to examine the effectiveness of the proposed technique.

- Models using proposed technique were compared with models using other state-of-the-art techniques.

- Models using proposed technique showed promising results.

- Proposed technique provides easier-to-understand explanations.

This paper is divided into five sections. Section one consists of reviews on the use of ML models in cancer prediction and ML techniques on interpretability. Section two delineates the methodology of the study, including data collection, data binarilization, feature selection, classification model, hyperparameter search, performance metrics, and model explanaation. Results and analyses are located in section three. Section four discusses the advantages of our proposed technique, contributions and limitations of this study, as well as future research plans. Finally, sections five concludes this paper by providing an overview of this study and future research directions.

## Materials and methods

In this section, we present our approach for cancer diagnosis using binarilized RNA-seq data. The approach consists of four parts: data binarilization, feature selection, model construction, and model explanation. First, raw RNA-seq data are binarlized. Then, univariate feature selection is applied to select the most relevant features from binarlized data. Processed data are then randomly splitted into a training set and a test set. The training set is used for both hyperparameter search and model training; whereas the test set is used for measuring the performance of trained models. The train-test process is repeated 10 times to get the average results. Models with the highest F1 scores is used to generate SHAP plots. And finally, SHapley Additive exPlanations is used to interpret trained models and determine most impactful features. Fig 1 demonstrates the flow of our proposed approach.

### Data collection

All data used in this study came from the National Cancer Institute Genomic Data Commons (GDC). $Log_2(x + 1)$ normalized illumina Hi-Seq RNA sequencing data was merged with clinical information based on their corresponding sample IDs. Samples without RNA-seq information were removed. Samples with primary, recurrent or metastatic tumor were considered as positive samples, while samples with solid tissue standard samples were considered as negative samples. In this study, only RNA-seq data were used, each of which contains the same 20,530 predictors. The properties of each dataset can be found in Table 1.

All four datasets were normalized using the min-max normalization technique before further processing. All positive samples were labeled as 1, whereas negatives were labeled as 0. After completing all processing, each dataset was divided into a training set and a test set using stratified split. The ratio between a training set and a test set was 80:20.

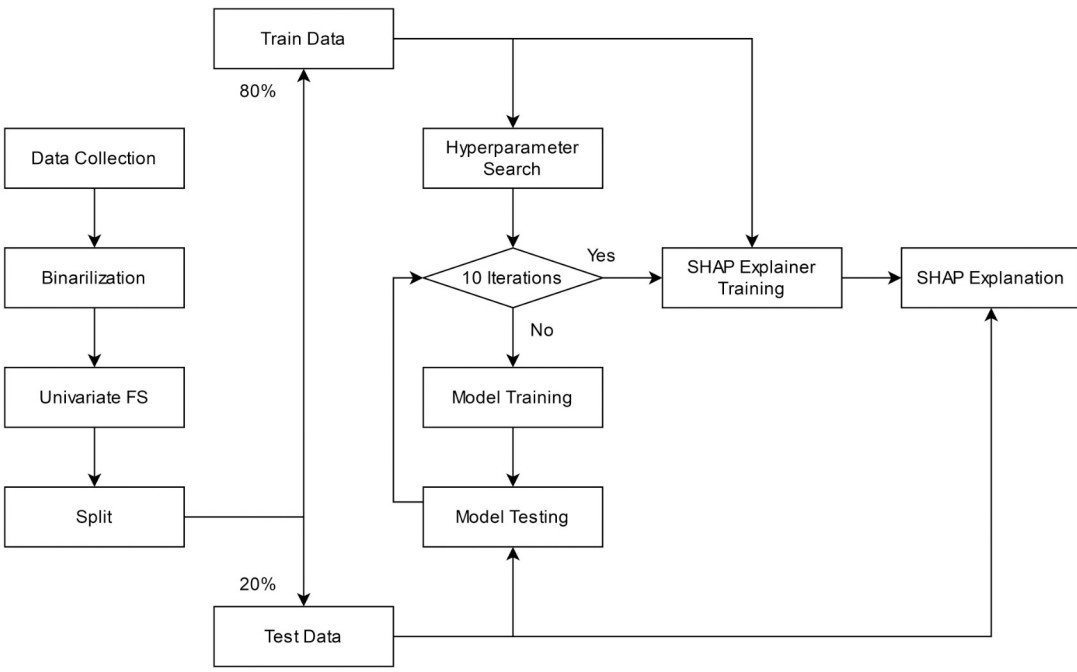

**Fig 1. Flowchart of the proposed approach.**

**Table 1. Dataset properties.**

| Name | Total | Positive | Negative |
|---|---|---|---|
| Liver | 422 | 372 | 50 |
| Lung | 1099 | 989 | 110 |
| Prostate | 550 | 498 | 52 |
| Thyroid | 572 | 513 | 59 |

## Data binarilization

Data binarilization can be seen as an extension of data discretization. Data discretization is the process of converting a feature of continuous values into a finite set of intervals, with each interval representing a range of original values. Although data discretization can reduce the complexity of a dataset, it does not make any feature more interpretable as each feature still contains more than two values. Data binarilization, on the other hand, creates a series of binary features for each continuous feature, with each binary feature representing a range of original values. For each set of binary features, each sample shall have only one positive value, representing the range of values the sample belongs to using the original continuous feature. Because of this characteristic, binarilization offers a more direct view than discretization on the relationships between features and outcomes generated by feature selection and model explanation methods.

A data binarilization tool can be constructed using Algorithm 1.

**Algorithm 1** Data Binarilization Algorithm

```
Require: Dataset D, I samples, J features and K binary features
    function DATA BINARILIZATION(D, K)
```

```
   BinarilizedDataset ← ∅
  for j ← 0, J do
    for k ← 0, K do
      TempSet ← ∅
      for i ← 0, I do
        if D_ij ≥ k/K ∧ D_ij < (k+1)/K then
          TempSet_i ← 1
        else
          TempSet_i ← 0
        end if
      end for
      BinarilizedDataset ← BinarilizedDataset ∪ TempSet
    end for
  end for
  return BinarilizedDataset
end function
```

## Feature selection

Because of the high dimensional nature of RNA-seq data and the additional dimensions created by the binarilization process, feature selection was used to remove irrelevant features to reduce overfitting and increase efficiency [34]. In this study, the Chi-Square test (Chi2) was selected. Chi2 is a statistical test primarily used to determine the level of dependency between two categorical variables. For each feature in the feature set, the corresponding $\chi^2$ value is calculated by Eq 1. All features are then ranked in descending order according to their corresponding $\chi^2$ values, with higher $\chi^2$ value indicating higher dependency between the feature and the label. Top-$k$ features with the highest $\chi^2$ values are selected to form the reduced feature set, where $k$ is the number of features chosen by the user. In this study, we chose 20, 200, 2000, 20000 features, representing 0.01%, 0.1%, 1%, and 10% of binarilized features.

$$\chi^2 = \sum_{i=1}^{m}\sum_{j=1}^{n}\frac{\left(O_{ij} - E_{ij}\right)^2}{E_{ij}} \tag{1}$$

In the above formula, $m$ denotes the number of features in the feature set, $n$ denotes the number of distinct labels, $O_{ij}$ denotes the observed frequency of having feature as $i$ and label as $j$, and $E_{ij}$ denotes the expected frequency of having feature as $i$ and label as $j$.

## Classification model

In order to examine the effectiveness of our proposed data processing technique, we chose to compare the performance of five models based on different algorithms, namely neural network, decision tree, random forest, XGBoost, and support vector machine.

**Neural network.** Neural Network (NN) essentially is just a stack of interconnected layers of nodes. The typical structure of a NN consists of an input layer, one or more than one hidden layers, and an output layer. Each node in the hidden and output layers is connected to all nodes in the previous layer [19]. Each connection has a associated weight. An activation function is also attached to each node in all but the input layer. The activation of an node is determined by the activation function, which calculates the activated value of that node base on all incoming connections. During the training process, the model adjusts all connections' weights to correctly predict the label of the input data. Sometimes, a dropout rate is also attached to each hidden layer to increase the generalizability of the model. Dropout is a process that reduces the overfitting problem by randomly deactivating some nodes in the hidden layers

during the training process. Comparing to other ML algorithms, NN has the advantage of being able to detect the non-linear relationships between input data and output labels. However, NN is computationally more expensive than other algorithms, which limits its applicable areas.

**Decision tree.** Decision tree (DT) is a powerful ML algorithm that is widely used [35]. A DT is made of two types of nodes and one-directional links. Each internal (non-leaf) node represents a test on an attribute, each link represents an outcome of the test, and each terminal (leaf) node represents a class label [36]. Comparing to other algorithms, it is more interpretable because of the tree-like structure, which can be easily converted to decision rules. The tree strcuture is also ideal for capturing interactions between features [37].

**Random forest.** The random forest (RF) algorithm is a decision tree-based bagging ensemble algorithm [38]. Comparing to other algorithms, RF is more robust against noise in data, has higher scalability, and offers strong performance in high-dimensional settings [39]. A random forest classifier generates multiple decision trees. Since not every tree uses all available features or samples, prediction made by each tree is different. Predictions from all trees are then collected and the label that most trees predict will be the final prediction. This process is called the majority voting process.

**XGBoost.** XGBoost is a decision tree-based boosting ensemble learning algorithm. Comparing to all previous boosting tree algorithms, XGBoost offers higher performance and efficiency [40]. Like RF, a XGBoost classifier also generates multiple decision trees. However, unlike bagging ensemble algorithms, boosting ensemble algorithms like XGBoost utilizes a iterative approach to train each sub model, then sequentially combine all sub models to form the final model. The weights of misclassified data in one model will be increased in the next model. Because of this iterative approach, boosting ensemble algorithms are more prone to the overfitting problem.

**Support vector machine.** Support vector machine is a classic ML algorithm for both linear and nonlinear data. In a classification problem, SVM searches for a decision boundary called hyperplane that classify input samples into one of two classes. To achieve this, SVM utilizes kernels to create a feature space in a higher dimension where linear separation is possible. Therefore, SVM is sensitive to kernel choice. SVM is considered highly accurate and less likely to overfit. However, SVM is less efficient than other algorithms [19].

## Hyperparameter search

In order to prevent overfitting and gain maximum performance, hyperparameter search was conducted for each model before training. Because of the lack of knowledge in determining the optimal values for some hyperparameters, random search was used in this study. In this study, each model had 200 variations; the value of each searched hyperparameter of each variation was picked randomly. Based on the F-Measure of each variation, the best performing one was chosen as the predictive model. Due to the small sample sizes of our datasets, 10-fold cross-validation was used in the hyperparameter search processes to avoid over/under-fitting [41]. Tables 2 to 6 contains the searched range for each hyperparameter for each algorithm.

## Evaluation metrics

After constructing and training the prediction models, we used the test sets to evaluate their performance. Because our datasets were imbalanced, metrics designed for imbalanced classification tasks. Specifically, we chose geometric mean (GMean), Matthews correlation coefficient (MCC), F-Measure (F1), and area under the receiver operating characteristic curve (AUC) as

**Table 2. NN hyperparameters search range.**

| Name | Value Range |
|---|---|
| Number of layers | 1, 2, 3 |
| Number of nodes per layer | 2—input size |
| Dropout rate | 0.1—0.3 |
| Activation Function | ReLU, ELU, GELU, Swish |
| Learning rate | 1 |
| Optimizer | Adadelta |

**Table 3. DT hyperparameters search range.**

| Name | Value Range |
|---|---|
| Criterion | gini, entropy |
| Max number of levels | 10—100 |
| Min samples split | 2—10 |
| Min samples leaf | 1—4 |
| Max features | sqrt, log2 |

**Table 4. RF hyperparameters search range.**

| Name | Value Range |
|---|---|
| Number of trees | 100—1000 |
| Criterion | gini, entropy |
| Max number of levels | 10—100 |
| Min samples split | 2—10 |
| Min samples leaf | 1—4 |
| Max features | sqrt, log2 |

**Table 5. XGBoost hyperparameters search range.**

| Name | Value Range |
|---|---|
| Number of trees | 100—1000 |
| Max number of levels | 10—100 |
| Learning rate | 0.01—0.1 |
| Gamma | 0.01—0.1 |
| Mininal child weight | 0.01—0.1 |
| Lambda | 0.01—0.1 |

**Table 6. SVM hyperparameters search range.**

| Name | Value Range |
|---|---|
| Kernel | Linear, Poly, RBF, Sigmoid |

**Table 7. Performance for liver data.**

| Model | Size | Acc | GMean | MCC | F1 | AUC |
|---|---|---|---|---|---|---|
| NN | 20 | 0.9824 | 0.9824 | 0.9166 | 0.9824 | 0.9683 |
| | 200 | 0.9847 | 0.9847 | 0.929 | 0.9847 | 0.9783 |
| | 2000 | 0.9812 | 0.9812 | 0.9094 | 0.9812 | 0.9503 |
| DT | 20 | 0.9694 | 0.9694 | 0.8487 | 0.9694 | 0.9133 |
| | 200 | 0.9624 | 0.9624 | 0.8119 | 0.9624 | 0.892 |
| | 2000 | 0.9518 | 0.9518 | 0.7581 | 0.9518 | 0.847 |
| RF | 20 | 0.9882 | 0.9882 | 0.9471 | 0.9882 | 0.9993 |
| | 200 | 0.9765 | 0.9765 | 0.8867 | 0.9765 | 0.9433 |
| | 2000 | 0.9647 | 0.9647 | 0.8237 | 0.9647 | 0.8933 |
| XGBoost | 20 | 0.9765 | 0.9765 | 0.8867 | 0.9765 | 0.9433 |
| | 200 | 0.9647 | 0.9647 | 0.8237 | 0.9647 | 0.8933 |
| | 2000 | 0.9647 | 0.9647 | 0.8237 | 0.9647 | 0.8933 |
| SVM | 20 | 0.9882 | 0.9882 | 0.9471 | 0.9882 | 0.9993 |
| | 200 | 0.9882 | 0.9882 | 0.9471 | 0.9882 | 0.9933 |
| | 2000 | 0.9765 | 0.9765 | 0.8867 | 0.9765 | 0.9433 |

our metrics [42–44]. We also included accuracy (ACC) as a metric although it is not particularly suitable for imbalanced classification tasks.

**Accuracy**: measures how many predictions are correct. It is calculated by dividing the number of correct predictions from the total number of predictions made, as shown in Eq 2.

$$Accuracy = \frac{T_p + T_n}{T_p + F_p + F_n + T_n} \tag{2}$$

**Table 8. Performance for lung data.**

| Model | Size | Acc | GMean | MCC | F1 | AUC |
|---|---|---|---|---|---|---|
| NN | 20 | 0.9912 | 0.9912 | 0.9527 | 0.9912 | 0.9951 |
| | 200 | 0.9951 | 0.9951 | 0.9735 | 0.9951 | 0.9973 |
| | 2000 | 0.992 | 0.992 | 0.9573 | 0.992 | 0.9956 |
| DT | 20 | 0.9947 | 0.9947 | 0.9713 | 0.9947 | 0.995 |
| | 200 | 0.9858 | 0.9858 | 0.9197 | 0.9858 | 0.9617 |
| | 2000 | 0.9814 | 0.9814 | 0.8936 | 0.9814 | 0.9451 |
| RF | 20 | 0.9956 | 0.9956 | 0.9756 | 0.9956 | 0.9975 |
| | 200 | 0.9912 | 0.9912 | 0.9527 | 0.9912 | 0.9951 |
| | 2000 | 0.9956 | 0.9956 | 0.9756 | 0.9956 | 0.9975 |
| XGBoost | 20 | 0.9912 | 0.9912 | 0.9527 | 0.9912 | 0.9951 |
| | 200 | 0.9912 | 0.9912 | 0.9496 | 0.9912 | 0.9748 |
| | 2000 | 0.9912 | 0.9912 | 0.9527 | 0.9912 | 0.9951 |
| SVM | 20 | 0.9867 | 0.9867 | 0.9312 | 0.9867 | 0.9926 |
| | 200 | 0.9956 | 0.9956 | 0.9756 | 0.9956 | 0.9975 |
| | 2000 | 1.0 | 1.0 | 1.0 | 1.0 | 1.0 |

**Table 9. Performance for prostate data.**

| Model | Size | Acc | GMean | MCC | F1 | AUC |
|---|---|---|---|---|---|---|
| NN | 20 | 0.9536 | 0.9536 | 0.7318 | 0.9536 | 0.8755 |
| | 200 | 0.9518 | 0.9518 | 0.7389 | 0.9518 | 0.9015 |
| | 2000 | 0.9591 | 0.9591 | 0.7436 | 0.9591 | 0.856 |
| DT | 20 | 0.9509 | 0.9509 | 0.7322 | 0.9509 | 0.883 |
| | 200 | 0.9364 | 0.9364 | 0.6393 | 0.9364 | 0.8345 |
| | 2000 | 0.9218 | 0.9218 | 0.5625 | 0.9218 | 0.786 |
| RF | 20 | 0.9909 | 0.9909 | 0.944 | 0.9909 | 0.95 |
| | 200 | 0.9545 | 0.9545 | 0.7132 | 0.9545 | 0.84 |
| | 2000 | 0.9727 | 0.9727 | 0.8286 | 0.9727 | 0.895 |
| XGBoost | 20 | 0.9636 | 0.9636 | 0.8023 | 0.9636 | 0.935 |
| | 200 | 0.9545 | 0.9545 | 0.7379 | 0.9545 | 0.885 |
| | 2000 | 0.9455 | 0.9455 | 0.6421 | 0.9455 | 0.79 |
| SVM | 20 | 0.9545 | 0.9545 | 0.7132 | 0.9545 | 0.84 |
| | 200 | 0.9636 | 0.9636 | 1.0 | 0.9636 | 0.845 |
| | 2000 | 0.9727 | 0.9727 | 0.8286 | 0.9474 | 0.895 |

**Sensitivity**: measures how well the positive class was predicted by calculating the positive rate, as shown in Eq 3.

$$Sensitivity = \frac{T_p}{T_p + F_n} \tag{3}$$

**Specificity**: measures how well the negative class was predicted by calculating the negative rate, as shown in Eq 4.

$$Specificity = \frac{T_n}{T_n + F_p} \tag{4}$$

**Table 10. Performance for thyroid data.**

| Model | Size | Acc | GMean | MCC | F1 | AUC |
|---|---|---|---|---|---|---|
| NN | 20 | 0.9817 | 0.9817 | 0.9039 | 0.9817 | 0.9567 |
| | 200 | 0.9965 | 0.9965 | 0.9815 | 0.9965 | 0.987 |
| | 2000 | 0.9965 | 0.9965 | 0.9818 | 0.9965 | 0.9907 |
| DT | 20 | 0.9696 | 0.9696 | 0.8344 | 0.9696 | 0.9057 |
| | 200 | 0.9409 | 0.9409 | 0.6492 | 0.9409 | 0.7903 |
| | 2000 | 0.9417 | 0.9417 | 0.6579 | 0.9417 | 0.7981 |
| RF | 20 | 0.9652 | 0.9652 | 0.8032 | 0.9652 | 0.8701 |
| | 200 | 0.9826 | 0.9826 | 0.9041 | 0.9826 | 0.9167 |
| | 2000 | 0.9652 | 0.9652 | 0.8011 | 0.9652 | 0.8333 |
| XGBoost | 20 | 0.9739 | 0.9739 | 0.8561 | 0.9739 | 0.9118 |
| | 200 | 0.9913 | 0.9913 | 0.9528 | 0.9913 | 0.9583 |
| | 2000 | 1.0 | 1.0 | 1.0 | 1.0 | 1.0 |
| SVM | 20 | 0.9913 | 0.9913 | 0.9561 | 0.9913 | 0.9951 |
| | 200 | 0.9739 | 0.9739 | 0.8561 | 0.9739 | 0.9118 |
| | 2000 | 0.9913 | 0.9913 | 0.9561 | 0.9913 | 0.9951 |

**Table 11. Performance for 20-feature liver data.**

| Model | Method | Acc | GMean | MCC | F1 | AUC |
|---|---|---|---|---|---|---|
| NN | AE | 0.9729 | 0.9729 | 0.865 | 0.9729 | 0.924 |
| | Chi2 | 0.98 | 0.98 | 0.9048 | 0.98 | 0.9583 |
| | PCA | 0.9871 | 0.9871 | 0.941 | 0.9871 | 0.9883 |
| | Hybrid | 0.9882 | 0.9882 | 0.9471 | 0.9882 | 0.9933 |
| | Proposed | 0.9824 | 0.9824 | 0.9166 | 0.9824 | 0.9683 |
| DT | AE | 0.9624 | 0.9624 | 0.8089 | 0.9624 | 0.8833 |
| | Chi2 | 0.9588 | 0.9588 | 0.7879 | 0.9588 | 0.8553 |
| | PCA | 0.9518 | 0.9518 | 0.7579 | 0.9518 | 0.847 |
| | Hybrid | 0.9694 | 0.9694 | 0.8511 | 0.9694 | 0.9263 |
| | Proposed | 0.9694 | 0.9694 | 0.8487 | 0.9694 | 0.9133 |
| RF | AE | 0.9647 | 0.9647 | 0.8237 | 0.9647 | 0.8933 |
| | Chi2 | 0.9647 | 0.9647 | 0.8237 | 0.9647 | 0.8933 |
| | PCA | 0.9882 | 0.9882 | 0.9424 | 0.9882 | 0.95 |
| | Hybrid | 0.9882 | 0.9882 | 0.9471 | 0.9882 | 0.9933 |
| | Proposed | 0.9882 | 0.9882 | 0.9471 | 0.9882 | 0.9993 |
| XGBoost | AE | 0.9529 | 0.9529 | 0.7577 | 0.9529 | 0.8433 |
| | Chi2 | 0.9647 | 0.9647 | 0.8237 | 0.9647 | 0.8933 |
| | PCA | 0.9882 | 0.9882 | 0.9471 | 0.9882 | 0.9933 |
| | Hybrid | 0.9882 | 0.9882 | 0.9471 | 0.9882 | 0.9933 |
| | Proposed | 0.9765 | 0.9765 | 0.8867 | 0.9765 | 0.9433 |
| SVM | AE | 0.9882 | 0.9882 | 0.9471 | 0.9882 | 0.9933 |
| | Chi2 | 0.9765 | 0.9765 | 0.8867 | 0.9765 | 0.9433 |
| | PCA | 0.9882 | 0.9882 | 0.9471 | 0.9882 | 0.9933 |
| | Hybrid | 0.9882 | 0.9882 | 0.9471 | 0.9882 | 0.9933 |
| | Proposed | 0.9882 | 0.9882 | 0.9471 | 0.9882 | 0.9993 |

**Geometric Mean**: is the squared root of the product of the sensitivity and specificity. It is calculated by Eq 5.

$$Geometric\ Mean = \sqrt{Sensitivity * Specificity} \tag{5}$$

**Matthews correlation coefficient**: calculates the *Pearson product − moment correlation coefficient* between correct and predicted values [10.1186/s12864-019-6413-7], as shown in Eq 6.

$$MCC = \frac{T_p * T_n - F_p * F_n}{\sqrt{(T_p + F_p) * (T_p + F_n) * (T_n + F_p) * (T_n + F_n)}} \tag{6}$$

**Precision**: measures the rate of correctly predicted positive samples, as shown in Eq 7.

$$Precision = \frac{T_p}{T_p + F_p} \tag{7}$$

**Recall**: is calculated the same way as sensitivity.

**Table 12. Performance for 20-feature lung data.**

| Model | Method | Acc | GMean | MCC | F1 | AUC |
|-------|--------|-----|-------|-----|-----|-----|
| NN | AE | 0.9991 | 0.9991 | 0.9953 | 0.9991 | 0.9995 |
| | Chi2 | 0.9894 | 0.9894 | 0.9403 | 0.9894 | 0.9738 |
| | PCA | 0.9889 | 0.9889 | 0.8928 | 0.9889 | 0.9493 |
| | Hybrid | 0.9982 | 0.9982 | 0.9901 | 0.9982 | 0.997 |
| | Proposed | 0.9536 | 0.9536 | 0.7318 | 0.9536 | 0.8755 |
| DT | AE | 0.9956 | 0.9956 | 0.9802 | 0.9956 | 0.994 |
| | Chi2 | 0.9845 | 0.9845 | 0.9151 | 0.9845 | 0.9691 |
| | PCA | 0.9881 | 0.9881 | 0.9361 | 0.9881 | 0.9832 |
| | Hybrid | 0.9898 | 0.9898 | 0.946 | 0.9898 | 0.9842 |
| | Proposed | 0.9947 | 0.9947 | 0.9713 | 0.9947 | 0.995 |
| RF | AE | 0.9956 | 0.9956 | 0.9756 | 0.9956 | 0.9975 |
| | Chi2 | 0.9867 | 0.9867 | 0.9262 | 0.9867 | 0.9724 |
| | PCA | 0.9956 | 0.9956 | 0.9756 | 0.9956 | 0.9975 |
| | Hybrid | 1.0 | 1.0 | 1.0 | 1.0 | 1.0 |
| | Proposed | 0.9956 | 0.9956 | 0.9756 | 0.9956 | 0.9975 |
| XGBoost | AE | 1.0 | 1.0 | 1.0 | 1.0 | 1.0 |
| | Chi2 | 0.9965 | 0.9965 | 0.9746 | 0.9965 | 0.9773 |
| | PCA | 0.9956 | 0.9956 | 0.9756 | 0.9956 | 0.9975 |
| | Hybrid | 0.9912 | 0.9912 | 0.9496 | 0.9912 | 0.9748 |
| | Proposed | 0.9912 | 0.9912 | 0.9527 | 0.9912 | 0.9951 |
| SVM | AE | 1.0 | 1.0 | 1.0 | 1.0 | 1.0 |
| | Chi2 | 0.9867 | 0.9867 | 0.9262 | 0.9867 | 0.9724 |
| | PCA | 0.9956 | 0.9956 | 0.9756 | 0.9956 | 0.9975 |
| | Hybrid | 0.9912 | 0.9912 | 0.9527 | 0.9912 | 0.9951 |
| | Proposed | 0.9867 | 0.9867 | 0.9312 | 0.9867 | 0.9926 |

**F-Measure**: is the harmonic mean of precision and recall. It is calculated by Eq 8.

$$F - Measure = \frac{2 * Precision * Recall}{Precision + Recall} \tag{8}$$

**Area under the receiver operating characteristic curve**: measures the entire area underneath the receiver operating characteristic curve. It can be calculated by Eq 9.

$$AUC = \frac{S_p + n_p(n_n + 1)/2}{n_p * n_n} \tag{9}$$

In the above formulas, $T_p$ denotes the number of correctly predicted positive samples, $T_n$ denotes the number of correctly predicted negative samples, $F_p$ denotes the number of incorrectly predicted positive samples, $F_n$ and denotes the number of incorrectly predicted negative samples. For AUC, $S_p$ denotes the sum of the ranks of all positive samples, whereas $n_p$ and $n_n$ denote the number of positive and negative samples respectively.

## Model explanation

Being a black box algorithm, interpreting a random forest is difficult due to the lack of built-in feature importance. To address this issue, we decided to use a popular post-hoc explainer called SHapley Additive exPlanations (SHAP) to explain the relevance of each feature. SHAP

**Table 13. Performance for 20-feature prostate data.**

| Model | Method | Acc | GMean | MCC | F1 | AUC |
|---|---|---|---|---|---|---|
| NN | AE | 0.9718 | 0.9718 | 0.8261 | 0.9718 | 0.899 |
| | Chi2 | 0.9436 | 0.9436 | 0.6647 | 0.9436 | 0.8385 |
| | PCA | 0.96 | 0.96 | 0.7604 | 0.96 | 0.8835 |
| | Hybrid | 0.9491 | 0.9491 | 0.7237 | 0.9491 | 0.8955 |
| | Proposed | 0.9491 | 0.9491 | 0.7187 | 0.9491 | 0.882 |
| DT | AE | 0.9409 | 0.9409 | 0.6395 | 0.9409 | 0.8145 |
| | Chi2 | 0.9082 | 0.9082 | 0.4919 | 0.9082 | 0.765 |
| | PCA | 0.9218 | 0.9218 | 0.5627 | 0.9218 | 0.795 |
| | Hybrid | 0.9145 | 0.9145 | 0.525 | 0.9145 | 0.7775 |
| | Proposed | 0.9509 | 0.9509 | 0.7322 | 0.9509 | 0.883 |
| RF | AE | 0.9636 | 0.9636 | 0.7638 | 0.9636 | 0.845 |
| | Chi2 | 0.9545 | 0.9545 | 0.7132 | 0.9545 | 0.84 |
| | PCA | 0.9636 | 0.9636 | 0.7638 | 0.9636 | 0.845 |
| | Hybrid | 0.9545 | 0.9545 | 0.7379 | 0.9545 | 0.885 |
| | Proposed | 0.9909 | 0.9909 | 0.944 | 0.9909 | 0.95 |
| XGBoost | AE | 0.9636 | 0.9636 | 0.78 | 0.9636 | 0.89 |
| | Chi2 | 0.9 | 0.9 | 0.5194 | 0.9 | 0.81 |
| | PCA | 0.9273 | 0.9273 | 0.4811 | 0.9273 | 0.69 |
| | Hybrid | 0.9364 | 0.9364 | 0.5653 | 0.9364 | 0.74 |
| | Proposed | 0.9636 | 0.9636 | 0.8023 | 0.9636 | 0.935 |
| SVM | AE | 0.9636 | 0.9636 | 0.78 | 0.9636 | 0.89 |
| | Chi2 | 0.9091 | 0.9091 | 0.45 | 0.9091 | 0.725 |
| | PCA | 0.9636 | 0.9636 | 0.78 | 0.9636 | 0.89 |
| | Hybrid | 0.9545 | 0.9545 | 0.7132 | 0.9545 | 0.84 |
| | Proposed | 0.9545 | 0.9545 | 0.7132 | 0.9545 | 0.84 |

is built on basis of game theory concepts, specifically the shapley value. Shapley values are based on the idea that the outcome of a prediction should determine the importance of each feature involved. SHAP utilizes this idea by constructing multiple models with the same set of hyperparameters and training data, but different sets of features. After models are created, the marginal contribution of each feature is calculated by finding the difference between 1) the difference between the prediction of a model with that feature and the average prediction and 2) the difference between the prediction of a model without that feature and the average prediction. Then, SHAP value for each feature is calculated by taking the average of all marginal contributions of that feature [25]. In this study, we chose SHAP because it is able to offer a globally consistent explanation [45]. Specifically, we chose to use TreeExplainer that is built to explain ensemble tree models [46].

## Experiments

### Environmental setup

Program built for this study ran on machines in the Sun Lab at the Penn State Harrisburg. The machines in the Sun Lab ran on Intel(R) Xeon(R) W-2245 CPU @ 3.90GHz. It provided a RAM of size 128 GB. Our study was implemented using Python 3.10.6. Imbalanced-learn 0.9.1 was used for the implementation of GMean metric. Scikit-learn 1.2.2 was used for implementing Chi-Square feature selection, hyperparameter search, three classifiers including decision

**Table 14. Performance for 20-feature thyroid data.**

| Model | Method | Acc | GMean | MCC | F1 | AUC |
|---|---|---|---|---|---|---|
| NN | AE | 0.987 | 0.987 | 0.9335 | 0.987 | 0.978 |
| | Chi2 | 0.9765 | 0.9765 | 0.8719 | 0.9765 | 0.9206 |
| | PCA | 0.9713 | 0.9713 | 0.8561 | 0.9713 | 0.9472 |
| | Hybrid | 0.9809 | 0.9809 | 0.8988 | 0.9809 | 0.9525 |
| | Proposed | 0.9817 | 0.9817 | 0.9039 | 0.9817 | 0.9567 |
| DT | AE | 0.9435 | 0.9435 | 0.6784 | 0.9435 | 0.8102 |
| | Chi2 | 0.9565 | 0.9565 | 0.7727 | 0.9565 | 0.8911 |
| | PCA | 0.9348 | 0.9348 | 0.6354 | 0.9348 | 0.7943 |
| | Hybrid | 0.9496 | 0.9496 | 0.7092 | 0.9496 | 0.8356 |
| | Proposed | 0.9696 | 0.9696 | 0.8344 | 0.9696 | 0.9057 |
| RF | AE | 0.9478 | 0.9478 | 0.6893 | 0.9478 | 0.7868 |
| | Chi2 | 0.9826 | 0.9826 | 0.907 | 0.9826 | 0.9535 |
| | PCA | 0.9565 | 0.9565 | 0.7478 | 0.9565 | 0.8285 |
| | Hybrid | 0.9826 | 0.9826 | 0.9041 | 0.9826 | 0.9167 |
| | Proposed | 0.9652 | 0.9652 | 0.8032 | 0.9652 | 0.8701 |
| XGBoost | AE | 0.9565 | 0.9565 | 0.7478 | 0.9565 | 0.8285 |
| | Chi2 | 0.9565 | 0.9565 | 0.7594 | 0.9565 | 0.8653 |
| | PCA | 0.9826 | 0.9826 | 0.9041 | 0.9826 | 0.9167 |
| | Hybrid | 0.9739 | 0.9739 | 0.8663 | 0.9739 | 0.9486 |
| | Proposed | 0.9739 | 0.9739 | 0.8561 | 0.9739 | 0.9118 |
| SVM | AE | 1.0 | 1.0 | 1.0 | 1.0 | 1.0 |
| | Chi2 | 0.9739 | 0.9739 | 0.8561 | 0.9739 | 0.9118 |
| | PCA | 0.9913 | 0.9913 | 0.9528 | 0.9913 | 0.9583 |
| | Hybrid | 0.9826 | 0.9826 | 0.907 | 0.9826 | 0.9535 |
| | Proposed | 0.9913 | 0.9913 | 0.9561 | 0.9913 | 0.9951 |

tree, random forest, and support vector machine, and all other metrics. Shap 0.41.0 was used for the implementation of the SHAP explainer. TensorFlow 2.12.0 was used for building the neural network classifier. And xgboost 1.7.3 was used for implementing the xgboost classifier.

## Performance of models

We compared the performance of models using our proposed data processing technique. To examine how feature selection will affect models that use binarilized data, we ran several experiments with different number of binary features selected. The results of all experiments can be found from Tables 7 to 10. By examining Tables 9 and 10, we can see that the performance of DT is slightly worse than other methods, which is to be expected as other methods theoretically should perform better than DT because of their complexity. Overall, the increase in number of features didn't affect the performance of models by a large margin.

We also compare the performance of models using our proposed data binarilization technique with other models based on other feature selection or dimensionality reduction techniques. By examining Tables 11 to 14, we can see that models using our proposed technique, despite having to rely on less features because of binarilization, perform about the same as models using other techniques. This proves that not only certain genes, but also some value ranges of certain genes are not relevant to cancer prediction.

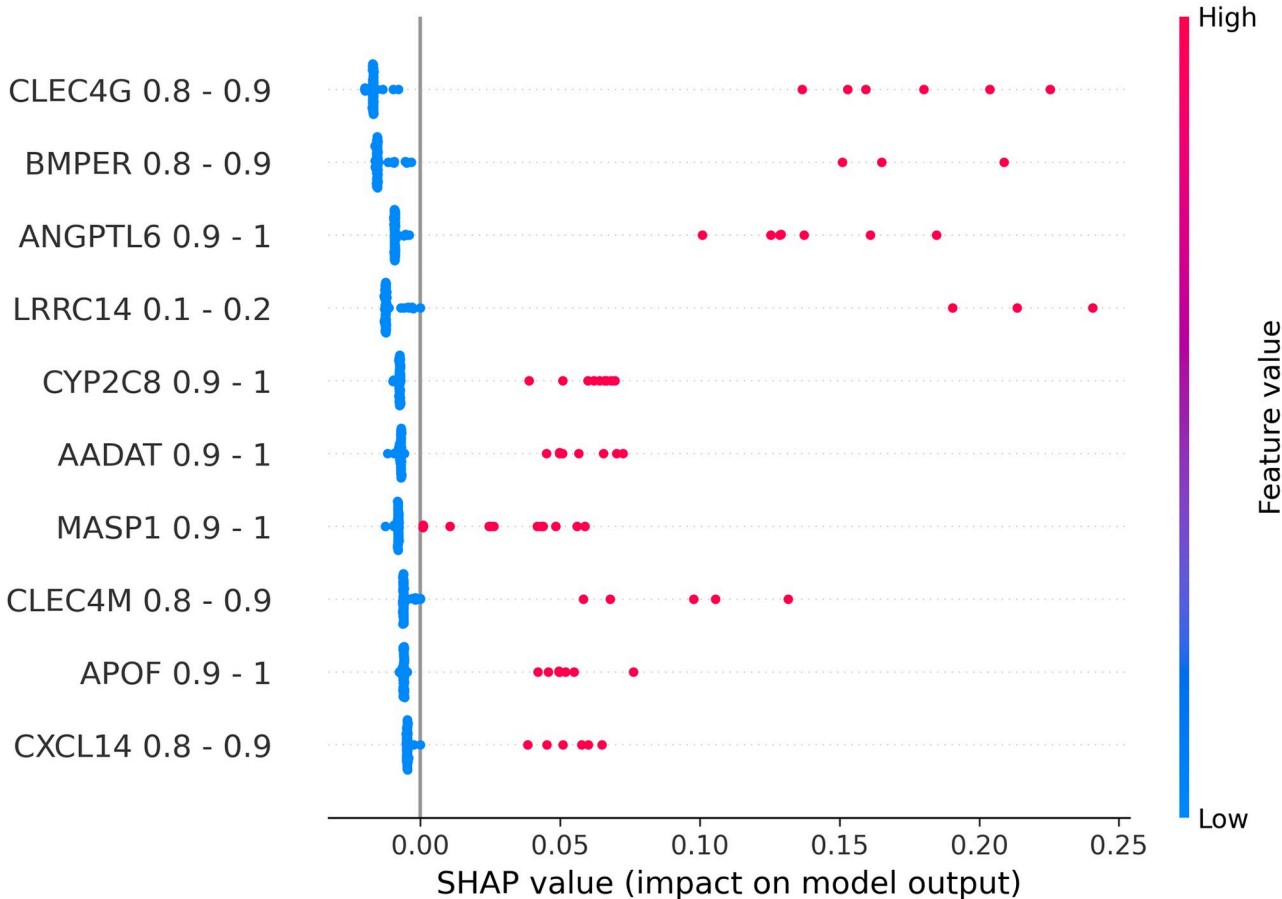

**Fig 2. Shap plots for 20-feature liver data.**

### Explanation results

To explain the relationship between input features and output labels, SHAP beeswarm plots were generated to visualize such relationships for each model. In a beeswarm plot, the feature names on the left side are ranked from top to bottom base on their corresponding absolute SHAP values. The horizontal axis represents the SHAP value of each data point. In places where multiple data points share the same SHAP value, dots are stacked vertically. The line at the 0 point separates samples that negatively contribute to the prediction from ones that have a positive contribution. We can see that, because our test data was imbalanced, the number of samples on the left side of the 0-line is significantly smaller than the one on the right side. The colorbar on the right side represents the value of the corresponding feature of each data point. Based on these properties, we can see that data binarilization gives a more direct view on the relationships between each gene and the predicted outcome by examining Figs 2–5, being able to show not only the most impactful features, but also their associated value ranges. Since each binary feature can only be either 0 or 1, SHAP plots for binarlized data only has two colors representing feature values.

### Discussion

In this study, we propose a novel data processing technique for cancer-related RNA-seq data. After binarilization, each gene is splitted into ten binary features. For each sample only one of

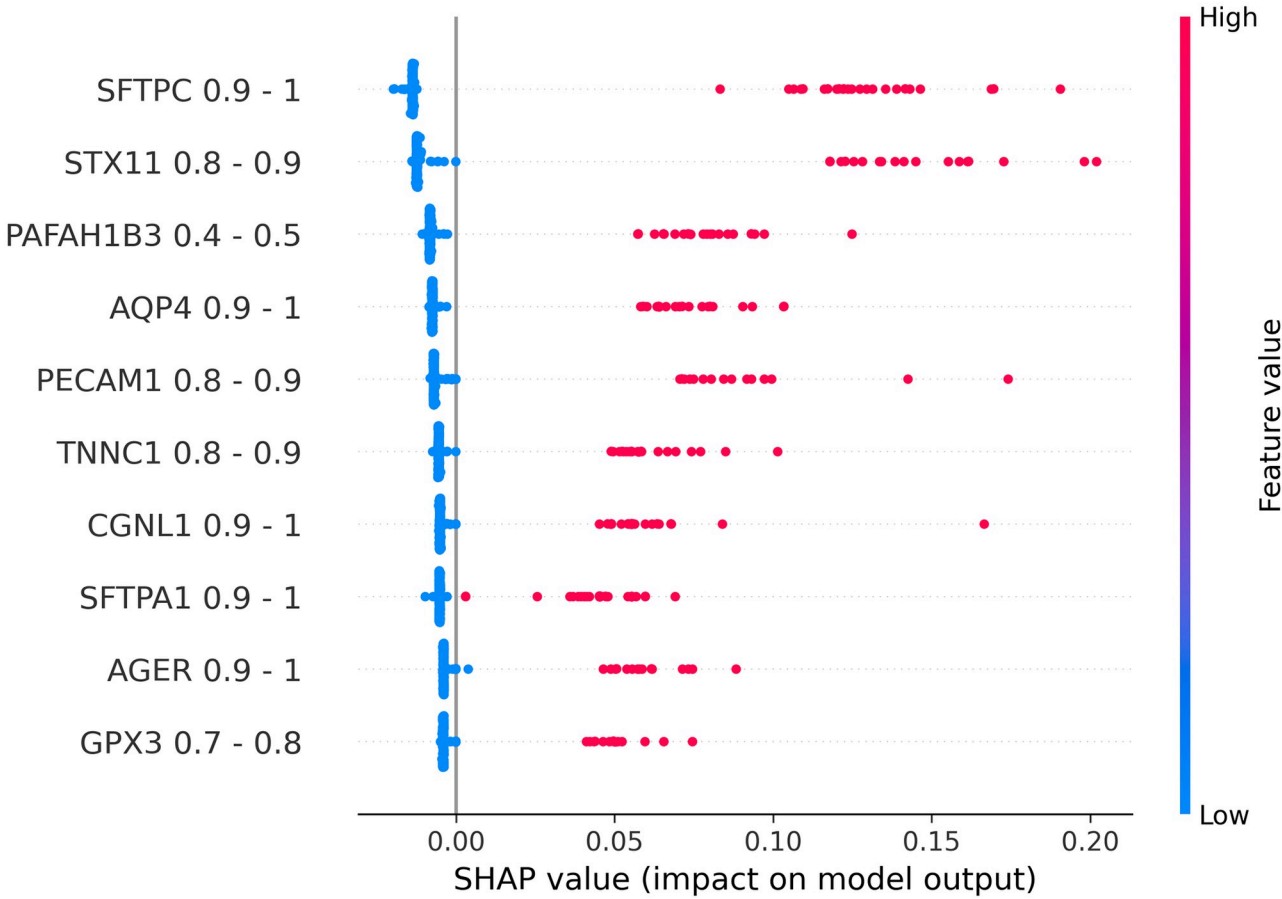

**Fig 3. Shap plots for 20-feature lung data.**

the ten binary features is positive, indicating the range of values the sample's original value of that feature lies between. Because of data binarization will increase the number of features, we performed feature selection to filter out irrelevant features. We compared the performance of models using binarilized features with models using continuous features. The results show that data binarization does not affect the predictive performance of models. For model explanation, we used SHAP to rank all features in terms of relevance to prediction. Comparing to other explanation models that use continuous data like iAFPs-EnC-GA, AIPs-SnTCN, and OncoNetExplainer, using binarlized data makes understanding results of SHAP analysis easier because the relevant features along with the relevant value range of the feature are revealed together.

Although we presented a novel approach that shows promising results, there are still some limitations to this study. First, the number of samples in each dataset was quite small. Thus, this work represents a proof-of-concept study for the data binarization approach. In future, we plan to apply this approach to larger datasets to examine its effectiveness. Second, the datasets were highly imbalanced. Although there are several imbalance treatment options like oversampling and undersampling, we decided not to use them as oversampling could create invalid samples that has more than one binary feature that belong to the same original feature be positive, whereas undersampling will further decrease the number of samples. Therefore,

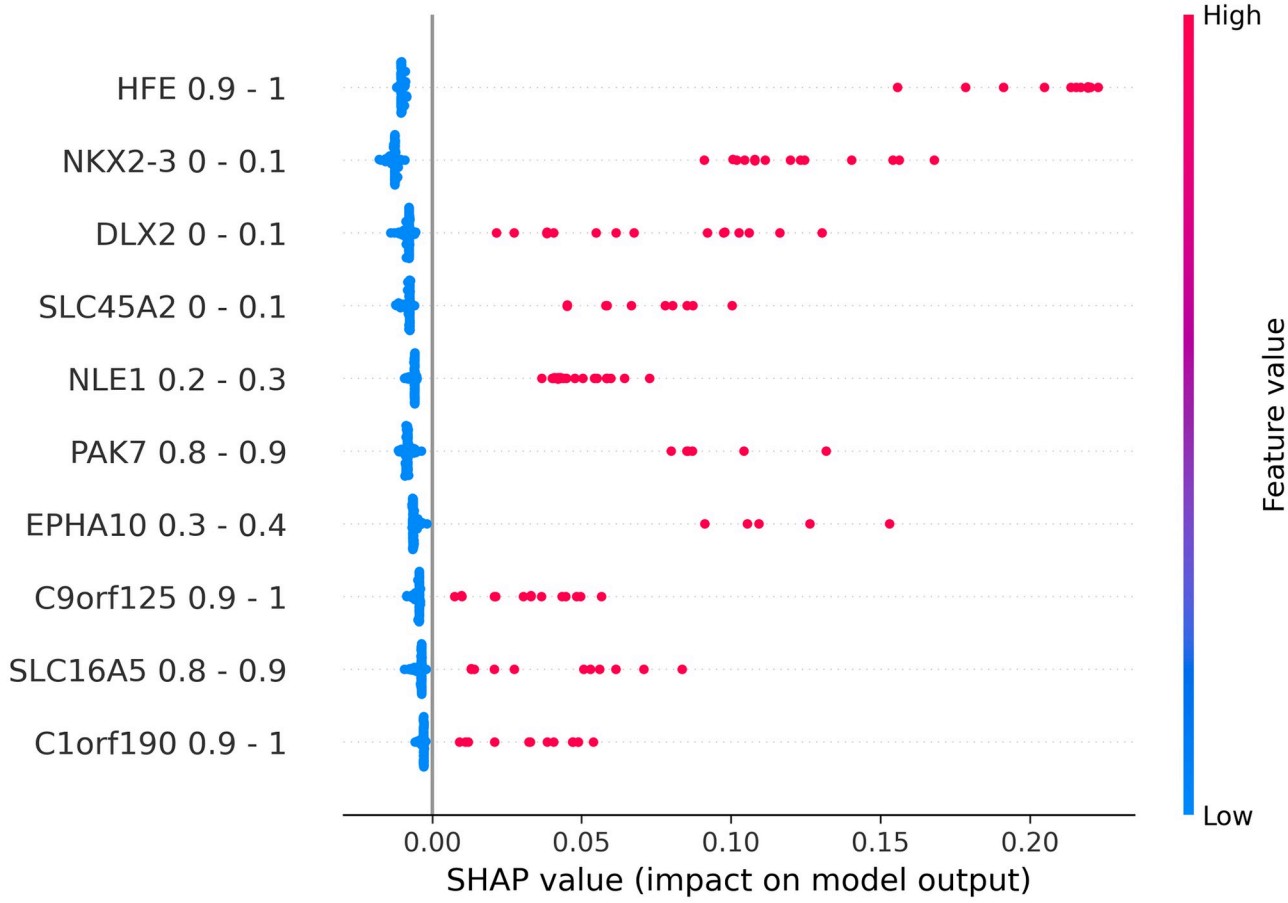

**Fig 4. Shap plots for 20-feature prostate data.**

we included metrics designed for imbalance classification tasks to mitigate this issue. Third, because some of the models included in this study were black box models, SHAP was used to provide explanations for them. However, post-hoc explanation techniques like SHAP is prone to biases exist in training data, which could lead to misleading and unfaithful explanations of models. Therefore, researchers suggest that white box models should be developed for fields involve high-stakes decision making [47]. Following such suggestions, we plan to utilize our proposed data processing technique to build more interpretable white box models in our future studies. Last but not least, the hidden properties of the datasets, such as the demography of patients and collection time, could have significant impact on the performance of data-driven techniques like ML models. We plan to collaborate with clinicians to test our proposed approach in the real world in our future studies.

## Conclusion

Early detection of cancer can increase patients' survival chances [48]. Recent developments in technology have enabled the use of new diagnostic methods like ML-assisted models. Because ML models excel at processing complex data, this allows researchers to utilize high-dimensional data like RNA-seq data to predict cancer patients and extract relevant biomarkers. However, ML models suffer from problems like poor interpretability. In this study, we proposed a

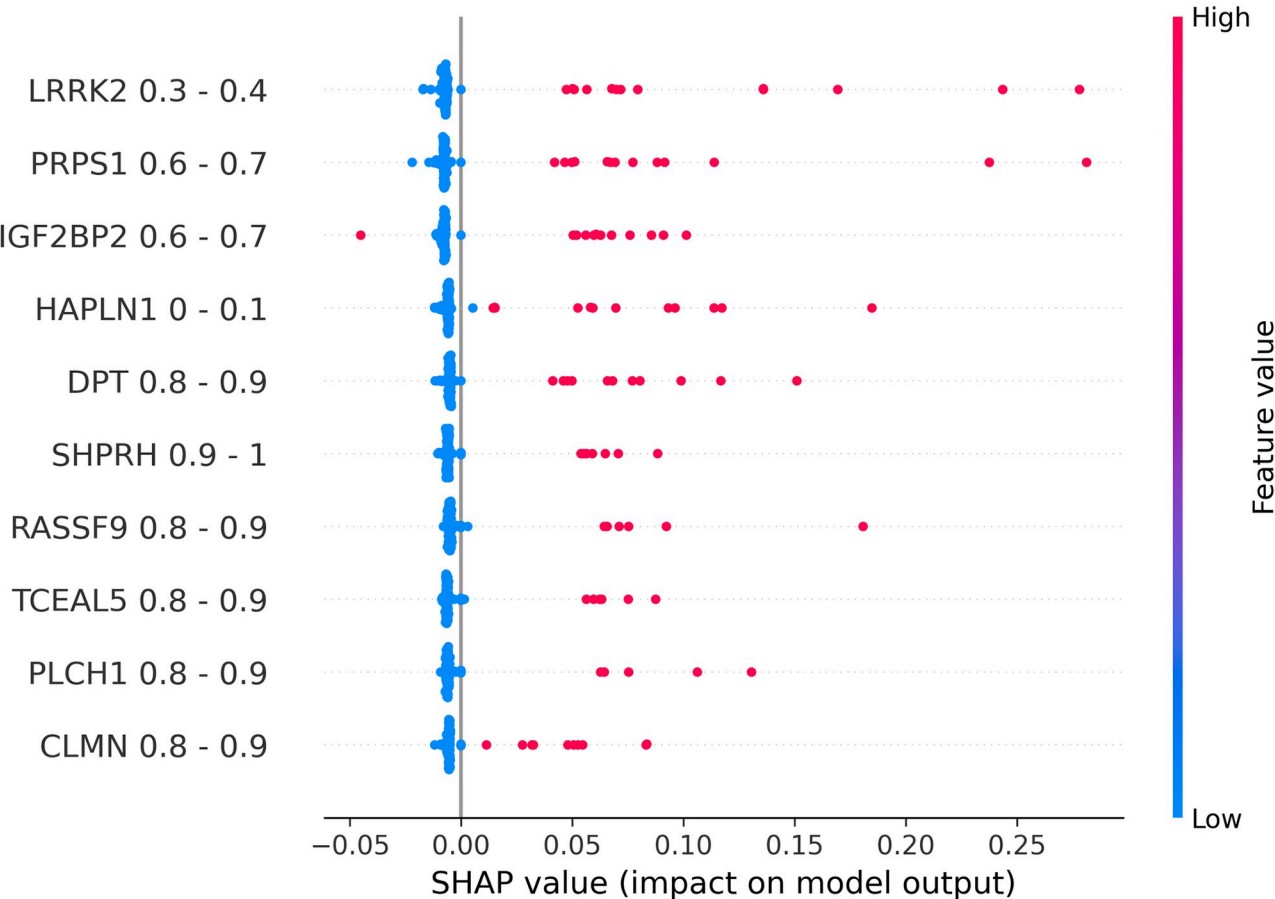

**Fig 5. Shap plots for 20-feature thyroid data.**

novel approach that utilize data binarilization to increase interpretability of ML models for cancer diagnosis. We proved that models using binarlized data can achieve the same level of performance while relying on less features. We also showed that data binarilization offers higher interpretability by offering a direct view on how each feature impacts the outcome. In future, we plan to address some limitations mentioned above by proposing new algorithms and establishing collaborations with clinicians. We also plan to apply this approach to address other healthcare problems.

## Author Contributions

**Conceptualization:** Tianjie Chen, Md Faisal Kabir.

**Data curation:** Tianjie Chen.

**Investigation:** Tianjie Chen, Md Faisal Kabir.

**Methodology:** Tianjie Chen.

**Supervision:** Md Faisal Kabir.

**Validation:** Tianjie Chen, Md Faisal Kabir.

**Writing – original draft:** Tianjie Chen.

**Writing – review & editing:** Md Faisal Kabir.

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
