## [Decision Letter · Decision Letter 0]

26 Feb 2024

PONE-D-23-33650Interpretable machine learning approach for cancer prediction through data binarilizationPLOS ONE

Dear Dr. Kabir,

Thank you for submitting your manuscript to PLOS ONE. After careful consideration, we feel that it has merit but does not fully meet PLOS ONE’s publication criteria as it currently stands. Therefore, we invite you to submit a revised version of the manuscript that addresses the points raised during the review process.

**ACADEMIC EDITOR: Major Revision**

We look forward to receiving your revised manuscript.

Kind regards,

Shahid Akbar, PhD

Academic Editor

PLOS ONE

Journal Requirements:

3. Please remove your figures from within your manuscript file, leaving only the individual TIFF/EPS image files, uploaded separately. These will be automatically included in the reviewers’ PDF.

Reviewers' comments:

Reviewer's Responses to Questions

**Comments to the Author**

1. Is the manuscript technically sound, and do the data support the conclusions?

Reviewer #1: Yes

Reviewer #2: Yes

2. Has the statistical analysis been performed appropriately and rigorously? 

Reviewer #1: Yes

Reviewer #2: Yes

3. Have the authors made all data underlying the findings in their manuscript fully available?

Reviewer #1: Yes

Reviewer #2: Yes

4. Is the manuscript presented in an intelligible fashion and written in standard English?

Reviewer #1: Yes

Reviewer #2: Yes

5. Review Comments to the Author

Reviewer #1: 1. At the end of introduction, the authors should add novelty and contributions in points.

2. in the related work section, the authors should discuss the peptide based approaches by citing the recent predictors such as, iACP-GAEnsC, cACP, cACP-2LFS, and cACP-DeepGram for the reader concerns.

3. the quality of figures are poor, authors should revised figures in 300dpi.

4. how the authors handle the overfitting issues of the proposed model.

5. in SHAP interpretation, the authors are advised to incorporate and discuss the recent predictors such as iAFPs-EnC-GA, and AIPs-SnTCN.

6. what should be the future directions of the proposed model.

Reviewer #2: 1. To validate the effectiveness of the proposed model, the authors needs a thorough comparison with existing state of the art methods.

2. What should be real life applications and importance of the proposed model.

3. For clear visualization, the authors should be represented in 300 dpi.

4. I suggest to incorporate a discussion section to shows the contribution, advantages, limitations, and improvement than existing studies.

5. The recent computational models needs to be cited related to biomedical engineering and ML such as; doi.org/10.3389/fgene.2020.539227, doi.org/10.1016/j.bbe.2020.05.010, and 10.1109/TNSRE.2016.2636367

6. PLOS authors have the option to publish the peer review history of their article (what does this mean?). If published, this will include your full peer review and any attached files.

Reviewer #1: No

Reviewer #2: No

---

## [Decision Letter · Decision Letter 1]

16 Apr 2024

Explainable machine learning approach for cancer prediction through binarilization of RNA sequencing data

PONE-D-23-33650R1

Dear Dr. Kabir,

We’re pleased to inform you that your manuscript has been judged scientifically suitable for publication and will be formally accepted for publication once it meets all outstanding technical requirements.

Kind regards,

Shahid Akbar, PhD

Academic Editor

PLOS ONE

Additional Editor Comments (optional):

Reviewers' comments:

Reviewer's Responses to Questions

**Comments to the Author**

1. If the authors have adequately addressed your comments raised in a previous round of review and you feel that this manuscript is now acceptable for publication, you may indicate that here to bypass the “Comments to the Author” section, enter your conflict of interest statement in the “Confidential to Editor” section, and submit your "Accept" recommendation.

Reviewer #1: All comments have been addressed

Reviewer #2: All comments have been addressed

2. Is the manuscript technically sound, and do the data support the conclusions?

Reviewer #1: Yes

Reviewer #2: Yes

3. Has the statistical analysis been performed appropriately and rigorously? 

Reviewer #1: Yes

Reviewer #2: Yes

4. Have the authors made all data underlying the findings in their manuscript fully available?

Reviewer #1: Yes

Reviewer #2: Yes

5. Is the manuscript presented in an intelligible fashion and written in standard English?

Reviewer #1: Yes

Reviewer #2: Yes

6. Review Comments to the Author

Reviewer #1: My comments are successfully addressed by the authors. therefore i suggest to accept the paper. i hope this paper this paper will perform a key role in research academia

Reviewer #2: the authors have successfully addressed all of my comments and i have no further comment. hence the paper can be accepted from my side.

7. PLOS authors have the option to publish the peer review history of their article (what does this mean?). If published, this will include your full peer review and any attached files.

Reviewer #1: No

Reviewer #2: No

---

## [Editor Report · Acceptance letter]

29 Apr 2024

PONE-D-23-33650R1 

PLOS ONE

Dear Dr. Kabir, 

I'm pleased to inform you that your manuscript has been deemed suitable for publication in PLOS ONE. Congratulations! Your manuscript is now being handed over to our production team.

Kind regards, 

on behalf of

Dr. Shahid Akbar 

Academic Editor

PLOS ONE